# A Wireless Body Sensor Network for Clinical Assessment of the Flexion-Relaxation Phenomenon

**Michele Paoletti** [1] **, Alberto Belli** [1] **, Lorenzo Palma** [1] **, Massimo Vallasciani** [2] **and Paola Pierleoni** [1,*]

[1] Department of Information Engineering (DII), Università Politecnica delle Marche, 60131 Ancona, Italy; m.paoletti@pm.univpm.it (M.P.); a.belli@univpm.it (A.B.); l.palma@pm.univpm.it (L.P.)

[2] Istituto di Riabilitazione Santo Stefano, 62018 Porto Potenza Picena, Italy; massimo.vallasciani@sstefano.it

[*] Correspondence: p.pierleoni@univpm.it; Tel.: +39-0712204847

**Abstract:** An accurate clinical assessment of the flexion-relaxation phenomenon on back muscles requires objective tools for the analysis of surface electromyography signals correlated with the real movement performed by the subject during the flexion-relaxation test. This paper deepens the evaluation of the flexion-relaxation phenomenon using a wireless body sensor network consisting of sEMG sensors in association with a wearable device that integrates accelerometer, gyroscope, and magnetometer. The raw data collected from the sensors during the flexion relaxation test are processed by an algorithm able to identify the phases of which the test is composed, provide an evaluation of the myoelectric activity and automatically detect the phenomenon presence/absence. The developed algorithm was used to process the data collected in an acquisition campaign conducted to evaluate the flexion-relaxation phenomenon on back muscles of subjects with and without Low Back Pain. The results have shown that the proposed method is significant for myoelectric silence detection and for clinical assessment of electromyography activity patterns.

**Keywords:** flexion-relaxation phenomenon; surface electromyography; wearable device; WBSN; automatic detection of the FRP

## 1. Introduction

The Flexion-Relaxation Phenomenon (FRP) term was adopted in 1955 by Floyd and Silver analysing the erector spinae muscles [1]. It consists of a back muscle electrical activity silence which typically occurs during the trunk full flexion. This effect is believed to be the result of ligaments activity and other passive elements of the spine that absorb the load of muscles. The erector spinae muscles (extensors of the trunk) contract when the trunk is flexed from the upright position acting as gravity antagonists. Floyd and Silver observed that myoelectric quiescence was caused by a reflex due to stretching in which the load torque of the upper body was transferred from the active to the passive spinal elements. It was also shown that, although the surface muscle activity was electrically very reduced, muscles continued to provide support through the stretching of the passive elements [2], and some of the deep muscles remaining electrically active in load support [3]. In the literature, it is known that the pain interferes with both afferent and efferent aspects of neuromuscular control [4–6]. Generally, in healthy subjects without Low Back Pain (LBP) stories, the FRP is statistically present, while in LBP patients the phenomenon is frequently absent. In order to evaluate if the subject has normal neuromuscular patterns amongst the various physiological indicators of LBP, the FRP has been one of the most studied surface electromyographic responses in the literature [1,3,7–9]. A lack of FRP was significant in pathological patients with pain, perceived disability, and re-injury fear. Furthermore, cases of healthy subjects without FRP, and LBP subjects with FRP (typically when the

pain is chronic) have been reported, but they were less frequent [10]. Sihvonen has reported that the FRP absence was more easily observed in subjects with LBP presence than patients without pain during the test [11]. Since the FRP absence is often used as an indicator for low back dysfunctions [12,13], several studies have used different methods to quantify the myoelectric activity, and discriminate the subjects, knowing a priori the health conditions [10,14–16]. The Flexion Relaxation Ratio (FRR) is one of these methods used to quantify FRP level and try to discriminate healthy subjects from LBP patients [11].

Some research works proposed in the literature have investigated the FRP phenomenon [17,18] and the evaluation methods to identify its presence/absence in healthy subjects [19,20]. An accurate method for the FRP analysis is the Visual Inspection Method (VIS), consisting of visually identify the phenomenon presence or absence by a subjective analysis of the processed sEMG signal, on the selected muscles [20]. This approach requires experience, it cannot be used by non-expert examiners and it causes a strong waste of time. However, thanks to its reliability, the VIS method was used to compare the algorithm performances to detect the FRP in healthy subjects [20]. Because muscular activities appear during the relaxation phase, in the case of LBP subjects, the automatic evaluation of the relaxation phase limits is a non-trivial problem for this kind of patients compared to healthy subjects [21] and automatic methods are needed to accurately quantify first a parameter associated with the FRP level (for example the FRR) and then identify FRP presence/absence. Alison et al. have used two types of VIS: VIS1 based on the raw EMG data and VIS2 based on the linear envelope of the EMG data [20]. However, the visual inspection only based on the sEMG signal is possible only when the patterns are recognizable through a visual analysis (typically, as in the case of Alison, when there are completely healthy subjects with regular patterns).

In the literature, several studies have proposed systems for the evaluation of FRP only on healthy subjects. In particular, Ritvanen et al. [22] have used one sEMG system while Alison Schinkel-Ivy et al. [18] have added a motion capture system to it. Sihvonen et al. [23] have proposed a similar system based on a wired Body Sensor Network. In this study, we wish to evaluate the performance of a Wireless Body Sensor Network (WBSN) able to analyse, quantitatively and objectively, the surface electromyography of the low back muscles (longissimus and multifidus) and automatically detect FRP presence/absence in both healthy subjects and patients with LBP. We focused attention on the quantification and evaluation of FRP rather than the discrimination in healthy and LBP subjects starting from the FRP obtained (without knowing the health conditions), which represents the next step of the analysis.

Generally, the FRP phenomenon is evaluated by observing the surface electrical activity of the spinal extensor muscles during a motion task where the subject reaches the maximal trunk flexion and returns to an upright position. This motion task, also known as flexion-relaxation test (or forward bend test), is mainly composed of four phases: standing, flexion, full flexion, and extension. In this test, starting from the upright position (Standing phase), the subject bends forward (Flexion phase). Once the bending of the torso reaches the maximum bending value of the trunk (Full-Flexion phase) naturally without straining the back, the subject returns to the initial position (Extension phase). During the execution, the lumbar spine is exploited in the first $50°$ of flexion phase, while in the remaining degrees the flexion occurs through the rotation of the pelvis [24]. In the extension phase instead inverse happens: rotation of the pelvis followed then by a lumbar spine extension. FRP statistically occurs in healthy subjects when the spine is about at $90°$, compared to the standing position, which is in the full flexion phase [25,26]. Therefore, an accurate FRP clinical assessment on back muscles requires a method for the surface electromyography signals analysis, especially in the full flexion phase. It is necessary, in order to identify each phase of the flexion-relaxation test, to process the data acquired by a system able to estimate the inclination of the subject during the bending movement. Motion analysis systems [27–29] and electronic goniometers [30] have been proposed for inclination detection during the flexion-relaxation movement. However, video-based analysis is time-consuming and limited to a given space under observation, while electronic goniometers are bulky and, therefore,

they can hinder the natural movement. In order to overcome these limitations, we have used a system based on non-obtrusive wearable and portable devices able to estimate the subject's inclination [31–33]. Such devices, compared to the electronic goniometers, have achieved an average error less of 4° for angle estimation and movement analysis [34].

Systems based on wearable devices have obtained similar performances when they were compared to optoelectronic systems (they represent the gold standard for motion analysis). As demonstrated in the literature, a comparative study has been proposed to validate a system composed of two wearable devices for lumbar inclination detection and it has shown an average error less of 2° compared to an optoelectronic system [35]. Despite the use of wearable devices was proposed and validated in gait analysis [36,37], there isn't currently a common agreement on which is the most appropriate approach to estimate the subject's inclination [38,39] and identify the different phases in the flexion-relaxation test to observe the electromyography signals and facilitate the FRP identification (with VIS and FRR methods).

In this paper, we propose a WBSN for the clinical assessment of the FRP during the flexion-relaxation test. It is composed of two separate systems: non-invasive wireless surface electromyography (sEMG) sensors in association with an inertial wearable device. With the algorithm developed by us, through a separate signal processing, subsequent synchronization, and overlap, we have obtained a single integrated network of sensors. Recent studies have proposed a system for assessment of the FRP even in LBP subjects. In particular, Ducina et al. have used a 12-camera motion analysis system to determine the inclination [40]. Other studies have started to use wearable wireless inertial systems to study the FRP [41,42]. However, these studies have not proposed automatic algorithms for detecting FRP or at least have not provided the results obtained with them. In this study, starting from the data acquired by the WBSN, an algorithm based on the FRR method was implemented to automatically detect the FRP and the obtained results have been compared with that deriving from the visual inspection. The developed software was tested using a dataset composed of healthy subjects (also called controls) and LBP subjects (also called patients) [43].

## 2. Materials and Methods

Given the lack of a standard procedure for clinical FRP assessment on back muscles, in this work, we have proposed a WBSN to automatically analyse the multichannel surface-electromyography signals together with the real movement performed by the subject. It consists of a wireless surface electromyography system composed of four sEMG sensors and one wearable device with triaxial accelerometer, gyroscope, and magnetometer sensors embedded. As shown in Figure 1, the wearable device was positioned on the first lumbar vertebra and the electrodes of the 4 sEMG sensors were placed along the fibers [44] of the longissimus left, longissimus right, multifidus left, multifidus right muscles, respectively. Longissimus muscles are part of the erector muscles of the vertebral column together with the iliocostalis and spinal muscles. The myoelectric signals of the longissimus muscles were acquired from two pairs of electrodes (LSX and LDX) positioned two fingers apart in a lateral direction from the spinous process L1 [45]. Multifidus muscles are part of the deep muscles of the trunk as they are in close contact with the spine. The myoelectric signals of the multifidus muscles were acquired from two pairs of electrodes (MSX and MDX) positioned on the line connecting the caudal tip of the posterior superior iliac spine to the space between L1 and L2, at the level of the spinous process of L5, 2–3 cm from the medial line [45]. The application surface was cleaned, before applying electrodes on the skin, with an alcohol swab, and the appropriate professional paste was applied to the electrodes in order to reduce the effects of the resistance provided by the skin and impurities [46,47].

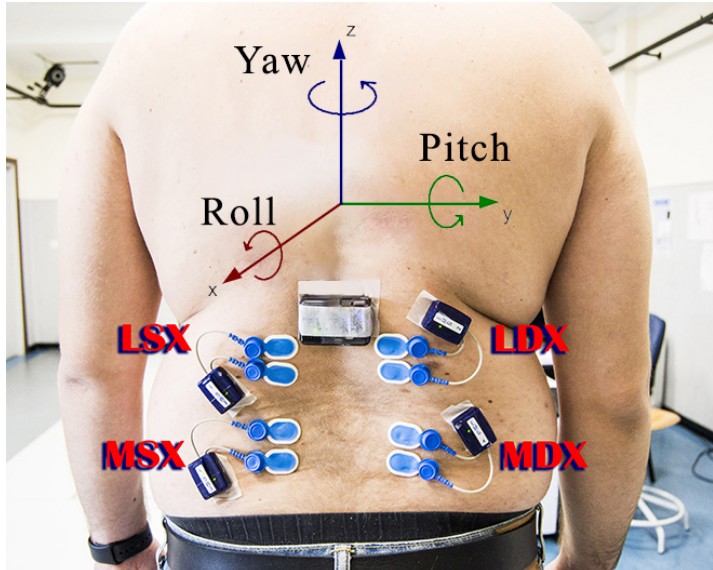

**Figure 1.** Positioning of the sEMG sensors and the wearable device on the subject under analysis. The electrodes were positioned following European recommendations for surface electromyography [45].

In order to identify muscles activity, the sEMG signals were acquired with a sampling frequency of 2000 Hz. The acquired sEMG signals were the following:

- Electromyography signal on left longissimus channel (LSX);
- Electromyography signal on right longissimus channel (LDX);
- Electromyography signal on left multifidus channel (MSX);
- Electromyography signal on right multifidus channel (MDX).

The wearable device signal, to estimate the inclination of the subject, was acquired with a sampling frequency of 128 Hz. The data collected by the wearable device were the following:

- Acceleration measured by the accelerometer (ACC);
- Angular velocity measured by the gyroscope (GYR);
- Magnetic field measured by the magnetometer (MAG).

The electromyography signals obtained by each sEMG sensor and the data collected by the wearable device were sent to a personal computer that acted as a central processing unit. The data acquired during the flexion-relaxation test, by the subjects under analysis, were processed and stored [12]. Before starting the forward bend test, the subject was placed with the arms on the side with the feet to the width of the shoulders, standing upright with the gaze straight and fixed on one point in order to avoid any artifact due to the alteration of the head position. During the flexion-relaxation test, the subject wore the proposed WBSN and he repeated 4 times a motion trial in which he was asked to naturally reach a bend angle about 90° without straining the lumbar region. One complete movement was called "cycle" and it was repeated 4 times (a compromise that allowed to have a good number of repetitions without exaggerating and stressing too much the muscles involved). As shown in Figure 2, each cycle consists of 4 phases listed below:

1. Standing—The subject keeps the standing position for about 4 s;
2. Flexion—The subject bend forward in order to naturally reach the full flexion position;
3. Full flexion—The subject keeps the full flexion position for about 4 s;
4. Extension—The subject return to standing position.

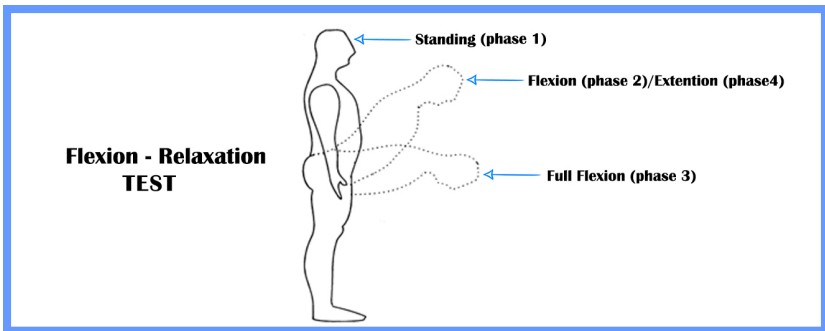

**Figure 2.** Representation of the movement performed by the subject during the flexion relaxation test.

According to McGorry et al. [12], the FRP may vary with changes in execution speed, prolonged static flexion, rest time, external load application. So, we must try to keep these parameters as constant as possible in order to make the results comparable over time.

The proposed WBSN was used in an extensive acquisition campaign conducted in collaboration with Santo Stefano Rehabilitation Institute (Porto Potenza Picena, Italy). The purpose was also to collect data from healthy and LBP subjects, to investigate the relationship between FRP and physical conditions. Before starting the acquisition campaign, demographic data, and patient history of each subject were registered to have a report about perceived pain and disability conditions. NRS-11 scale was used to identify the perceived pain during, before, and after the flexion-relaxation test execution [43,48]. Another patient condition measure (disability) was evaluated with the backill questionnaire [49] that assesses the ability to make or not a series of activities [43]. Procedures and experimental design of the acquisition campaign have been described in our previous study together with the complete dataset used to evaluate the FRP detection performances of the developed algorithm [43]. The dataset includes information and signals acquired on a total of 25 subjects submitted to the flexion-relaxation test, using the proposed WBSN.

## 3. Algorithm for FRP Clinical Assessment

An accurate FRP clinical assessment requires the analysis of the electromyography signals on back muscles and the measurement of the subject's inclination during the flexion-relaxation test. The WBSN automatically provides a clinical evaluation of the FRP through the algorithm proposed in this paper and described in the block diagram of Figure 3.

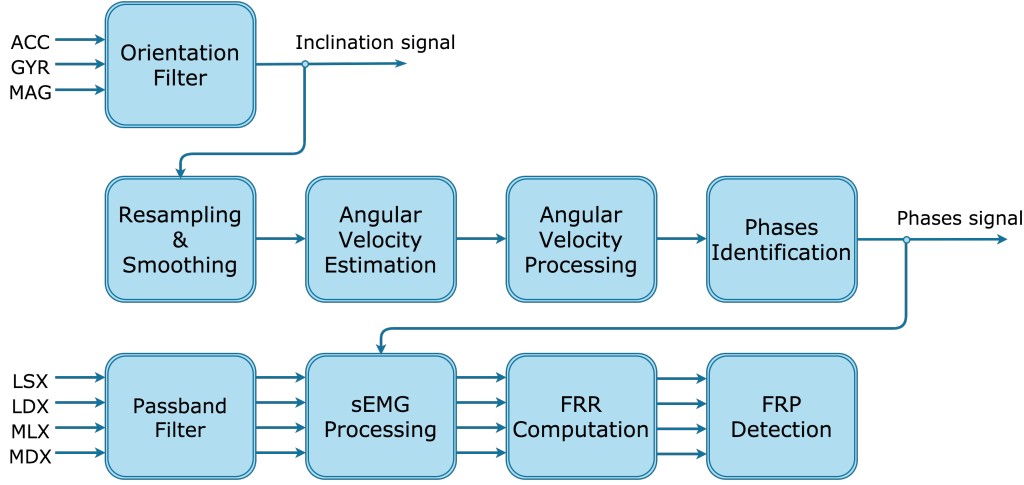

**Figure 3.** Block diagram of the proposed algorithm for clinical assessment of the FRP

Starting from the raw data acquired by the wearable device, an orientation filter [50,51] was implemented to provide the orientation of the subject's trunk in terms of Yaw, Pitch, and Roll angles [52].

As shown in Figure 1, Yaw, Pitch, and Roll angles describe the rotations around the Z, Y and X axes, respectively. Such representation describes the actual orientation of the subject's trunk starting from a fixed initial frame and the pitch angle identifies the inclination signal used to be associated with each electromyography signal acquired from the back muscles. Processing the inclination data it is possible to obtain a signal, called "phases signal", to define the onset/end phase of the motion, on the flexion-extension test. The main steps used by the proposed algorithm to compute the phases signal are described in Figure 3, and an example of the inclination signal processing is shown in Figure 4. Due to the different sampling frequency compared to the sEMG sensors system, the inclination signal was resampled to 2 kHz using a linear interpolation function. This increase in frequency of the inclination signal, albeit fictitious through oversampling, is not significant due the sampling frequency used is high enough to fully describe the specific movement. Subsequently, the resampled inclination signal was processed using a moving average filter with a span equal to 100 (orange signal in Figure 4). The derivative of the previous signal respect the time ($d\theta/dt$) was made in order to compute the angular velocity and its absolute value (blue signal in Figure 4). To obtain a signal that was zero when there were variations and changes during the static phases the reciprocal of the absolute value of angular velocity was computed and subsequently filtered using a 3th-order one-dimensional median filter and a time moving average filter with a span equal to 100 (violet signal in Figure 4). This signal, called processed angular velocity, was compared with an empirical threshold level equal to 0.09, to produce the phases signal (green signal in Figure 4). This threshold value was empirically obtained by carrying out a series of tests. In particular, several acquisitions on healthy subjects with normal sEMG patterns were analysed by superimposing the "Inclination signal" on the "Processed angular velocity signal". The chosen threshold represented the best value able to identify the various phases since the sEMG patterns with that value coincided perfectly with the duration of the phases. When the processed angular velocity was greater than the threshold level, the phases signal was set to a high value to identify standing and full-flexion phases. Otherwise, when the processed angular velocity was under the threshold level the phases signal was set to a low value to identify flexion and extension phases. Thus, the proposed algorithm is able to identify: phase onset/end, cycle onset/end, flexion-extension test onset/end, discarding the samples of the signals that exceed the last phase of the last cycle.

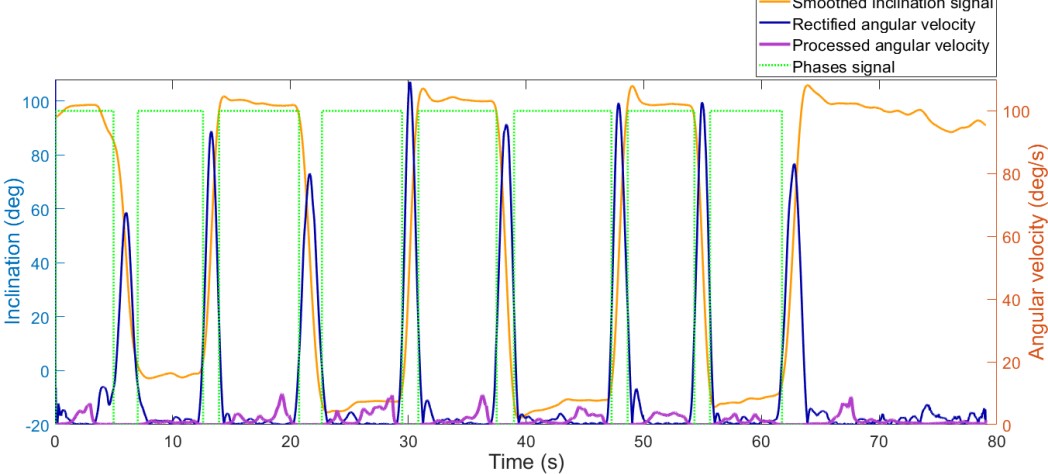

**Figure 4.** Signal processing of the inclination signal in order to obtain the "phases signal" which automatically defines the phases and cycles during a flexion-relaxation test.

In order to allow the evaluation of the myoelectric activity on back muscles, the sEMG signals were filtered using a sixth-order Butterworth passband filter 30 Hz ÷ 450 Hz. The best value to

filter ECG artifacts in sEMG signals [53] was 30 Hz, and 450 Hz was used to remove high-frequency harmonics [21,54]. The filtered sEMG signals (where signal exceed of the test end was discarded), the inclination, and the phases signal were superimposed providing the appropriate data to carry out the FRP analysis, as reported in the block diagram of Figure 3. Figure 5a–d shows the filtered sEMG signals, inclination signal, and phases signal processed by the proposed algorithm (they are referred to a control subject without LBP). Using this graphic representation, qualified medical personnel can easily carry out a visual inspection (VIS method) to analyse the back muscles activity and identify the presence/absence of FRP in each cycle of each sEMG channel. The greatest interest phase to make the decision, using the VIS method, is the full-flexion phase (number 3), where it's possible to observe FRP presence/absence.

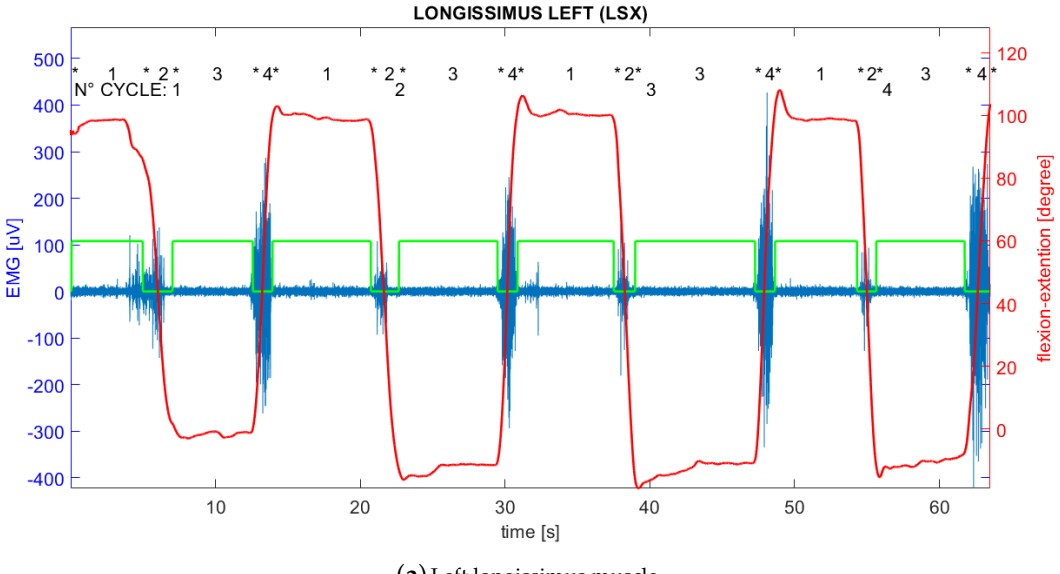

(**a**) Left longissimus muscle

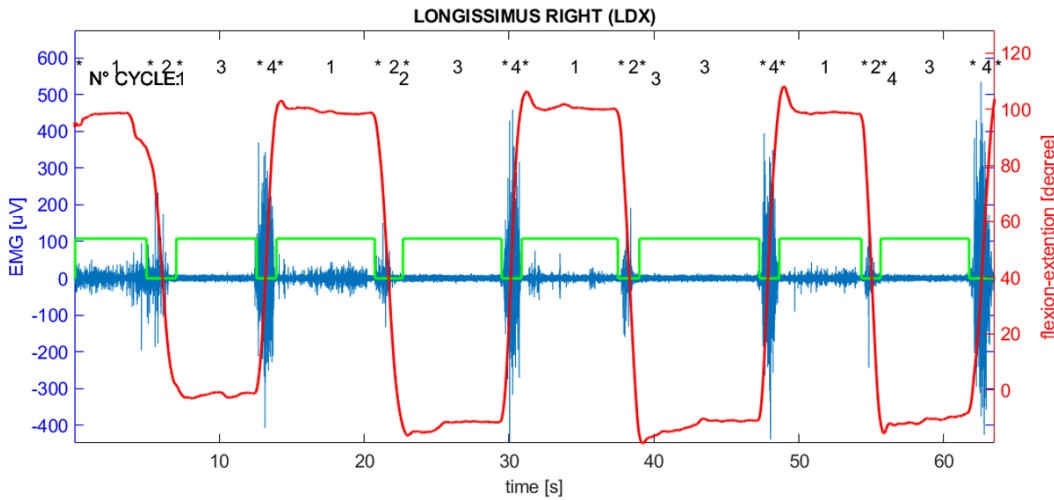

(**b**) Right longissimus muscle

**Figure 5.** *Cont.*

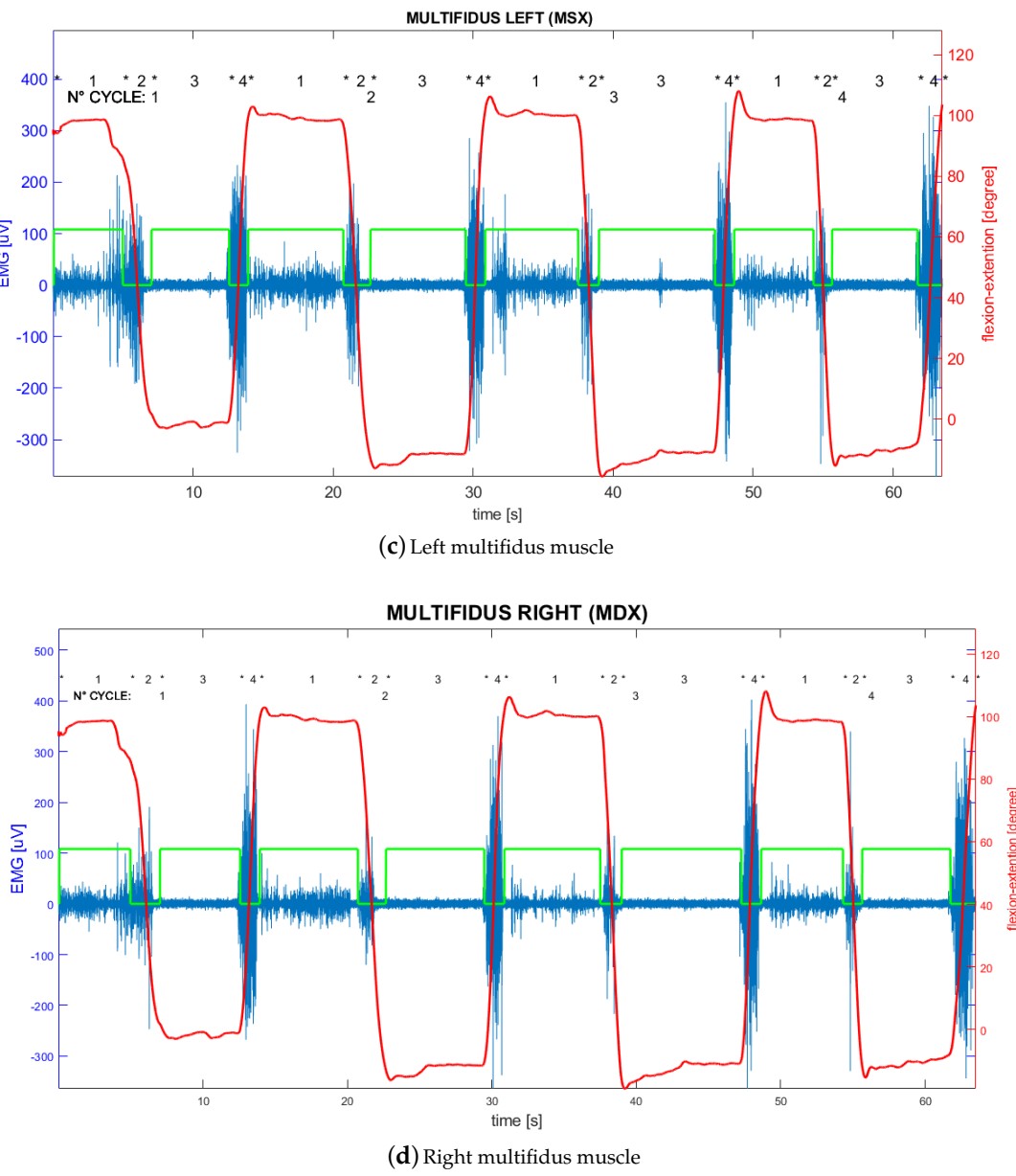

(**c**) Left multifidus muscle

(**d**) Right multifidus muscle

**Figure 5.** Graphic representation with the signals superimposition (filtered sEMG signal in blue, inclination signal in red, phases signal in green), phases (upper numbers) and cycles (lower numbers). It is referred to a healthy subject.

The second part of the algorithm consists to calculate the flexion-relaxation ratio (FRR) which is another technique, compared to the VIS method, to identify the FRP. The main difference is that the decision about FRP presence/absence in the VIS method is made by the doctor while using the FRR the final decision is made automatically by an algorithm. The goal was therefore to automate the decision-making process, to provide an objective contribution that could help the doctor in less time. Most of the FRRs used in the literature are the ratio of the EMG processed during the full-flexion phase on the numerator and EMG processed during the extension phase on the denominator. The reason is that theoretically under normal conditions, with healthy subjects, in the complete flexion we should obtain the lowest average activity thanks to FRP presence, while the average activity is higher, compared to all the other phases, during the extension (as is possible to see in Figure 6a–d); therefore, choosing a relationship between these two phases theoretically we have a greater gap (is easier to

make the decision). For example, in the same conditions, if we choose the ratio between the full flexion and the standing phase the average gap is less (as is possible to see in Figure 6a–d).

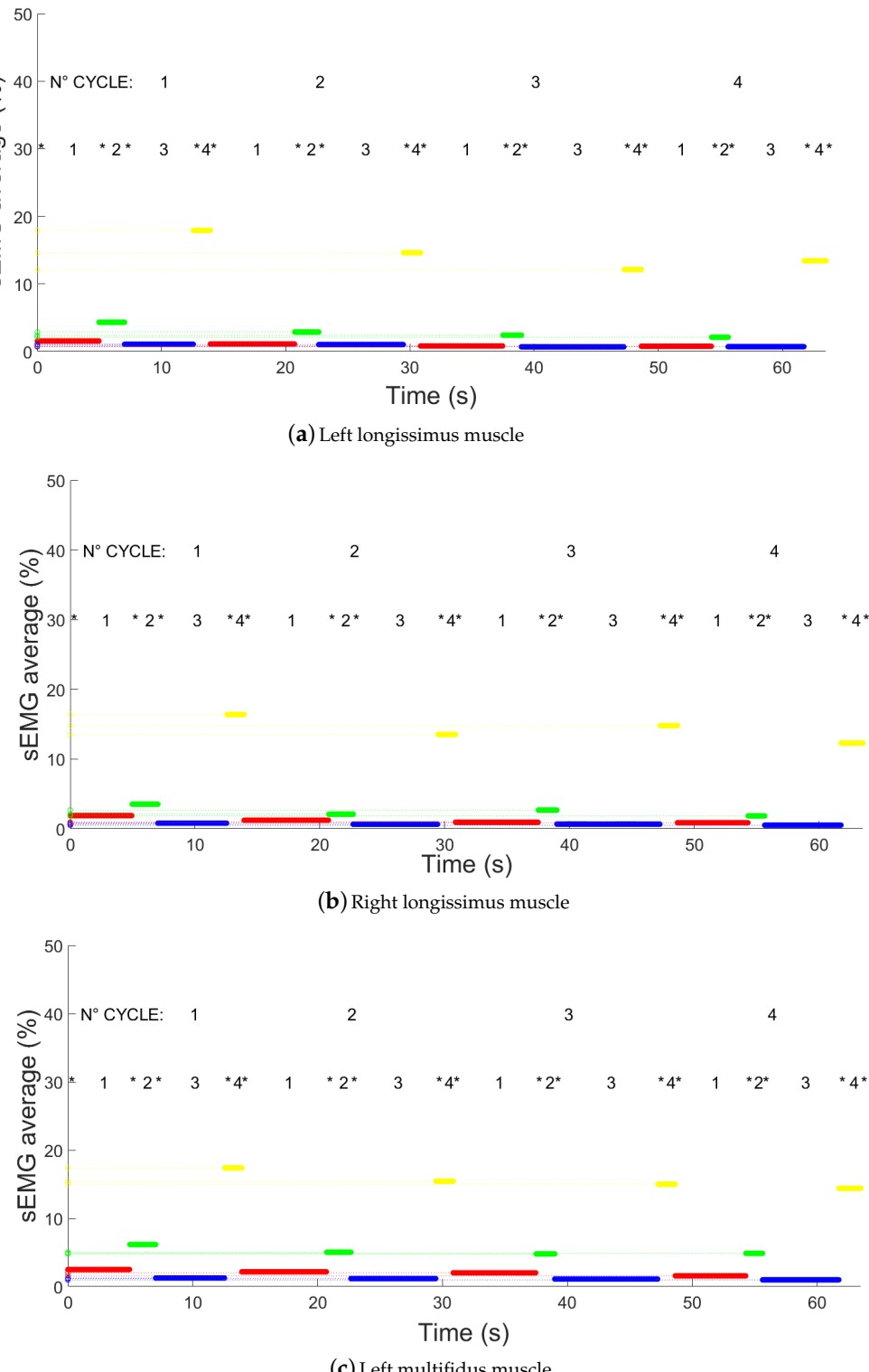

(**a**) Left longissimus muscle

(**b**) Right longissimus muscle

(**c**) Left multifidus muscle

**Figure 6.** *Cont.*

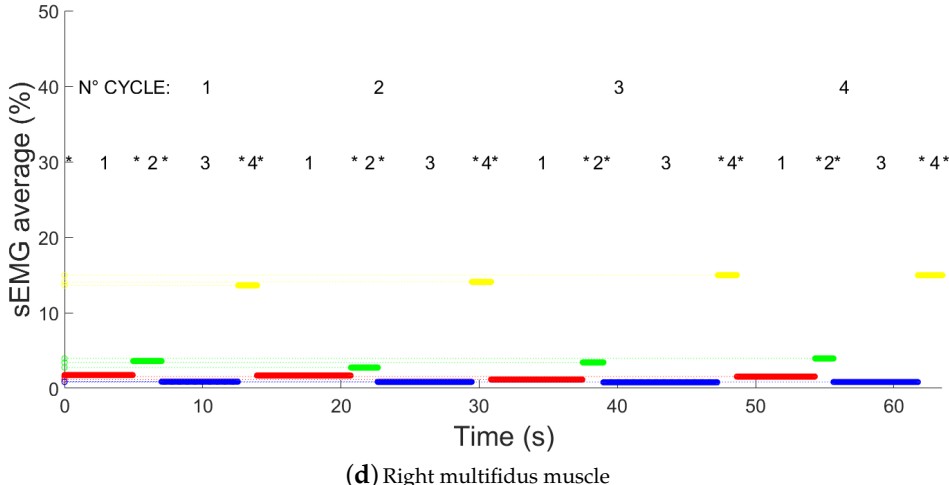

**(d)** Right multifidus muscle

**Figure 6.** sEMG filtered and rectified is normalized respect the max value of each cycle and the average sEMG levels, for each phase of each muscle, are expressed in percentage. Each phase is represented by a different colour: standing phase (red), flexion phase (green), full-flexion phase (blue), extension phase (yellow). They are referred to the same healthy subject of the previous graphs.

Calculate sEMG amplitudes ratios, between the motion phases, is a technique that allows normalization for repeated measures over time or for between-subject comparisons [12]. Ritvanenen et al. have computed the FRR by the ratio between the maximal RMS activity during 1 s of the flexion with the maximal RMS activity during 1 s of the full flexion; then the data were normalized by dividing them for the average sEMG activity during the standing phase [22]. Fernandes et al. have computed the FRR dividing the maximum RMS of EMG activity level during the flexion by the lowest mean EMG activity as measured over a 1-second interval during the full flexion phase [55]. In the literature there are many different types of FRRs, some of which have been explained and compared by Alison [20]. However, these studies did not report cutoff values to discriminate the presence/absence of FRP. Moreover, the comparison between FRR proposed in these studies is difficult due to the different factors used to evaluate the FRP and the lack of standardization. In order to overcome these limitations in this paper we propose an FRR method that uses a nominal threshold reference value to detect the presence or absence of FRP; this cutoff is an empirical value obtained during the analysis of the data collected in the acquisition campaign [43].

The specific FRR used in this study, also called flexion-extension ratio (FER), is the ratio between the average of the filtered and rectified sEMG signal during the full-flexion phase and the average of the filtered and rectified sEMG signal during the extension phase. This processing is made in the "FRR computation" block of Figure 3. The filtered sEMG signals were rectified and normalized respect to the max sEMG value in each cycle. Figure 7a–d shows the rectified and normalized sEMG signals where it is also illustrated the percentage angle (compared with the max angle) at which a new phase begins. These graphs are useful to evaluate the myoelectric activity on the back muscles and emphasizing the most common patterns: low activity in the standing phase, the activity increases in the flexion phase, myoelectric silence in full-flexion phase, and then the activity increases much more in the extension phase. As mentioned above, typically the myoelectric activity during the extension phase is greater than myoelectric activity during the flexion phase because it is necessary to contrast the torso strength, generated by the gravity acceleration, in opposition to the force generated by the muscles when they return to the starting position.

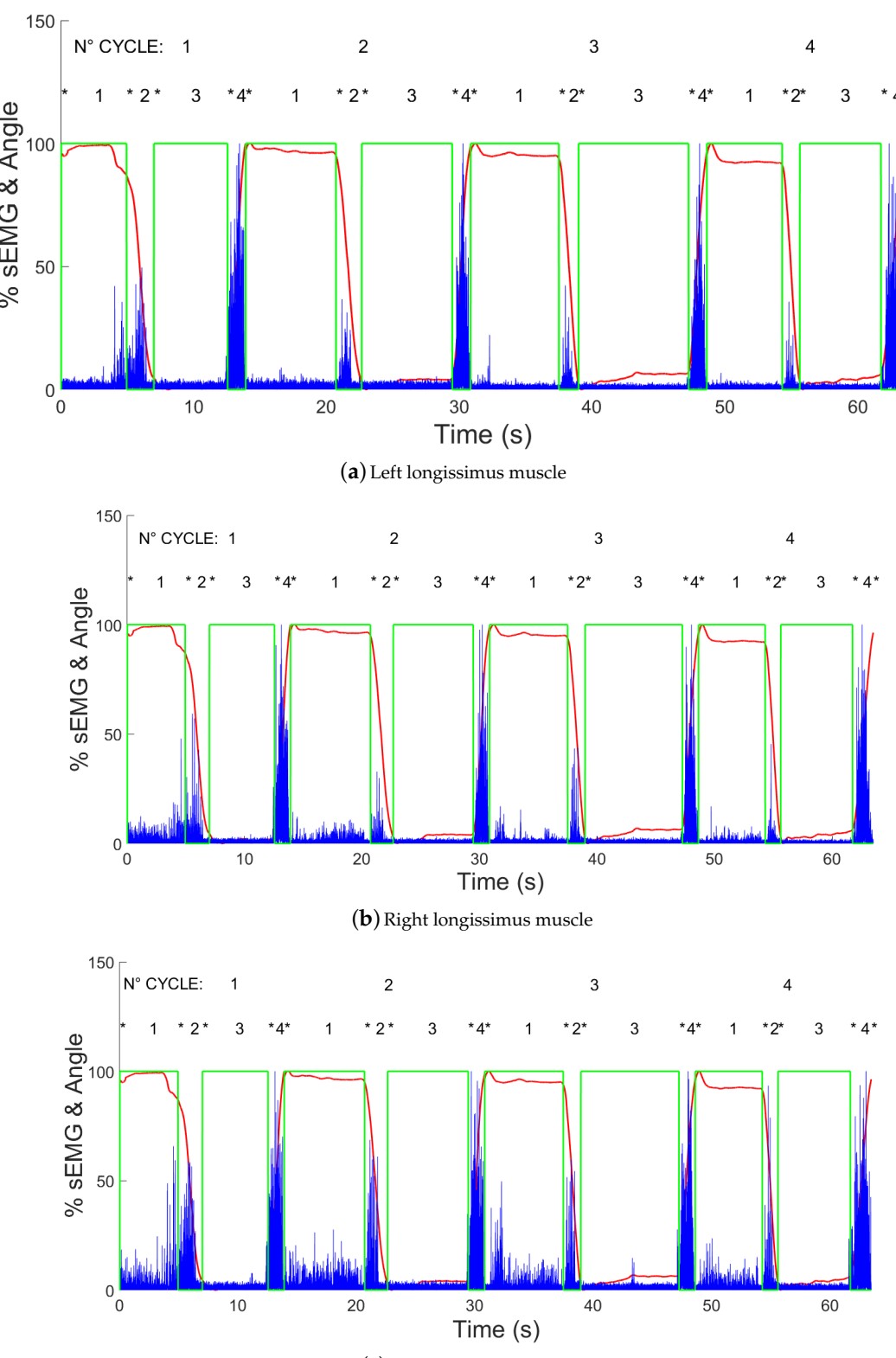

(**a**) Left longissimus muscle

(**b**) Right longissimus muscle

(**c**) Left multifidus muscle

**Figure 7.** *Cont.*

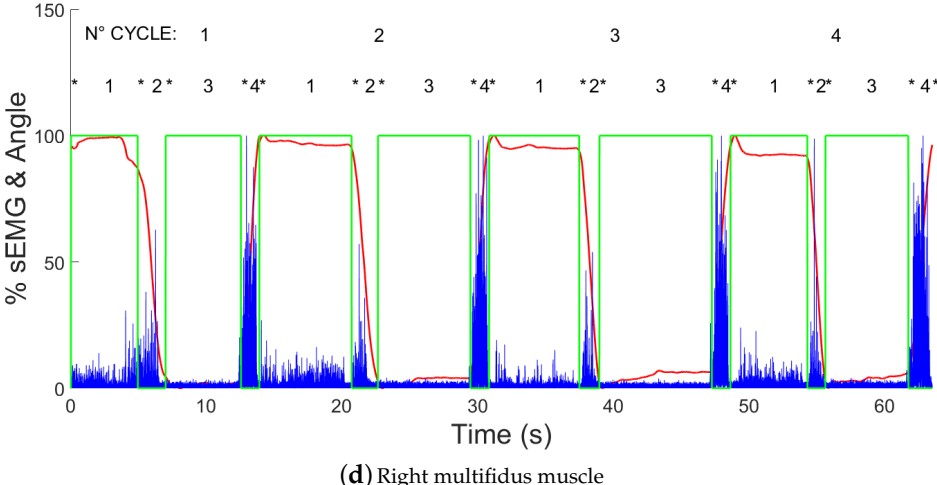

(**d**) Right multifidus muscle

**Figure 7.** Myoelectric activity for each phase and each muscle. Blue signal represents the sEMG normalized respect the max value of the cycle and it is expressed in percentage terms. The red signal is the inclination signal normalized respect the max value in the cycle and it is expressed in percentage terms. The green graph is the phases signal. They are referred to the same healthy subject of the previous graphs.

The processed sEMG signals were used to compute the FRR applying the following equation:

$$FRR_i^C = \frac{[\sum_{j=1}^{n}(|sEMG_j|/max(sEMG^i(t))) * 100]/n}{[\sum_{k=1}^{m}(|sEMG_k|/max(sEMG^i(t))) * 100]/m} \quad For\ i = 1,...,4 \quad\quad (1)$$

where:

- $i$ = $i$-th cycle;
- $j$ = $j$-th sample in full-flexion phase;
- $k$ = $k$-th sample in extension phase;
- $n$ = total samples in full-flexion phase;
- $m$ = total samples in extension phase;
- $sEMG^i(t)$ = represents the total signal filtered and synchronized in the $i$-th cycle, where $t$ is a discrete-time variable multiple of the sample time;
- $sEMG_j$ = represents the $j$-th amplitude of the sEMG signal (filtered and synchronized) in full flexion phase;
- $sEMG_k$ = represents the $k$-th amplitude of the sEMG signal (filtered and synchronized) in extension phase;
- $C$ = type of Channel (LSX, LDX, MSX, MDX).

For each cycle, in Equation (1) the numerator is represented with the blue signal while the denominator is represented with the yellow signal, as reported in Figure 6a–d.

It should happen that when the FRP is present the full flexion activity is very low respect extension activity (ratio near zero), while when the FRP is absent the full flexion activity approaches the extension activity level (ratio near one). The last algorithm step consists to compare each $FRR_i^C$ with a nominal threshold reference value $FRR^{Threshold}$ set at 0.35. So, the final stage (detection block in Figure 3) considers the FRP presence if the calculated value is below 0.35 ($FRR_i^C < FRR^{Threshold}$) while it considers the FRP absence if the ratio is equal or greater than 0.35 ($FRR_i^C \geq FRR^{Threshold}$). The $FRR^{Threshold}$ was empirically estimated and we can define it as *"the best value that reduces better the differences between the VIS method and FRR method"*.

## 4. Results

The algorithm was tested using the dataset published in our previous study [43]. It contains acquisitions of 25 volunteer subjects; they repeated four times the cycle as defined in the flexion-relaxation test procedure [43]. In each cycle the electromyography signals of four muscles were acquired. Therefore, a total number of 400 events were extracted for the evaluation of the performances using the proposed algorithm for FRP clinical assessment. To identify if the automatic algorithm was taking the correct decision, in terms of FRP identification, a comparison between FRR and VIS method results were presented. VIS method was taken as a benchmark because of, due to its accurate performances, it was commonly adopted in clinical and research applications [20]. The recordings collected in the dataset were evaluated by independent blind teams composed of medical experts. Using VIS method (on the sEMG signals with the superimposed inclination signal), the blind teams provided handwritten reports in which the occurrences of FRP in each cycle have been reported. The results of the VIS method were obtained by summarizing the handwritten reports and in cases of disagreement between the three blind teams, a final decision was reached by the majority. The VIS method results are shown in Table 1, where each event of the dataset is classified by a Positive (P) or a Negative (N) outcome. In events with the positive outcome the blind teams have ascertained the FRP presence in the cycle under examination while in the events with a negative outcome the FRP absence was identified. The VIS method is based on criteria found in the literature: "A clear, sudden reduction in motor activity" [20].

**Table 1.** Results of the VIS method. For each subject, the cycles of each channel are classified as Positive (P) or Negative (N) events.

| Subject ID | SEX | AGE | GROUP | LSX | | | | LDX | | | | MSX | | | | MDX | | | |
|---|---|---|---|---|---|---|---|---|---|---|---|---|---|---|---|---|---|---|---|
| | | | | 1 | 2 | 3 | 4 | 1 | 2 | 3 | 4 | 1 | 2 | 3 | 4 | 1 | 2 | 3 | 4 |
| 1 | F | 51 | LBP | P | P | P | P | P | P | P | P | P | P | P | P | P | P | P | P |
| 2 | F | 40 | HEALTHY | P | P | P | P | P | P | P | P | P | P | P | P | P | P | P | P |
| 3 | F | 34 | HEALTHY | P | P | P | P | P | P | P | P | P | P | P | P | P | P | P | P |
| 4 | M | 57 | LBP | N | P | P | P | N | P | P | P | N | N | N | N | N | N | P | P |
| 5 | M | 30 | LBP | N | N | N | N | N | N | N | N | N | N | N | N | N | N | N | N |
| 6 | M | 31 | HEALTHY | N | N | N | N | N | N | N | N | N | N | N | N | N | N | N | N |
| 7 | M | 35 | HEALTHY | P | P | P | P | P | P | P | P | N | P | P | P | P | P | P | P |
| 8 | M | 25 | HEALTHY | P | P | P | P | P | P | P | P | P | P | P | P | P | P | P | P |
| 9 | M | 58 | LBP | N | P | P | P | N | P | P | P | N | N | N | P | N | N | N | N |
| 10 | F | 52 | LBP | N | N | N | N | N | N | N | N | N | N | N | N | N | N | N | N |
| 11 | F | 46 | LBP | N | N | N | N | N | N | N | N | N | N | N | N | N | N | N | N |
| 12 | F | 40 | HEALTHY | P | P | P | P | P | P | P | P | P | P | P | P | P | P | P | P |
| 13 | M | 49 | LBP | N | N | N | N | P | P | P | P | N | N | N | N | N | N | N | N |
| 14 | F | 49 | LBP | P | P | P | P | P | P | P | P | N | N | N | N | N | N | N | N |
| 15 | F | 51 | LBP | N | P | P | P | N | P | P | P | N | N | N | N | N | N | N | N |
| 16 | F | 60 | HEALTHY | N | N | N | N | N | N | N | N | N | N | N | N | N | N | N | N |
| 17 | F | 36 | HEALTHY | P | P | P | P | P | P | P | P | P | P | P | P | P | P | P | P |
| 18 | M | 22 | HEALTHY | P | P | P | P | P | P | P | P | P | P | P | P | P | P | P | P |
| 19 | M | 52 | LBP | N | N | N | N | N | N | N | N | N | N | N | N | N | N | N | N |
| 20 | F | 22 | HEALTHY | P | P | P | P | P | P | P | P | P | P | P | P | P | P | P | P |
| 21 | M | 60 | HEALTHY | P | P | P | P | P | P | P | P | P | P | P | P | P | P | P | P |
| 22 | F | 51 | HEALTHY | N | N | N | N | N | N | N | N | N | N | N | N | N | N | N | N |
| 23 | M | 60 | LBP | N | N | N | N | N | N | N | N | N | N | N | N | N | N | N | N |
| 24 | M | 61 | LBP | N | N | N | N | N | N | N | N | N | N | N | N | N | N | P | P |
| 25 | M | 52 | HEALTHY | P | P | P | P | N | N | N | N | N | N | N | N | N | N | N | N |

An ideal algorithm, for FRP detection, should identify the FRP onset in all the events with a positive outcome and it should recognize FRP absence in all the events with a negative outcome. Comparing the results of the VIS method with those obtained by the proposed automatic algorithm for FRP detection is possible to identify four types of events classification:

- True Positive (TP)—The algorithm correctly reported FRP presence in an event with a positive outcome;
- False Positive (FP)—The algorithm incorrectly reported FRP presence in an event with a negative outcome;
- True Negative (TN)—The algorithm correctly reported FRP absence in an event with a negative outcome;
- False Negative (FN)—The algorithm incorrectly reported FRP absence in an event with a positive outcome.

The information, collected in the dataset, was processed by the proposed algorithm for FRP detection and the results have been shown together with the flexion-extension ratio values calculated in each cycle and channel (Table 2), using Equation (1). Table 3 shows the mean and the SD obtained using the algorithm for computing ratios between full-flexion and extension phase.

Since the number of dataset events was statistically significant, it was possible to derive accuracy, sensitivity, and specificity about the performances of the proposed algorithm using the following equations:

$$A_c = \frac{TP + TN}{P + N} = \frac{TP + TN}{TP + FP + TN + FN} = \frac{382}{400} = 95.5\% \tag{2}$$

$$S_e = \frac{TP}{P} = \frac{TP}{TP + FN} = \frac{195}{195 + 3} = 98.5\% \tag{3}$$

$$S_p = \frac{TN}{N} = \frac{TN}{TN + FP} = \frac{187}{187 + 15} = 92.6\% \tag{4}$$

**Table 2.** Results of the FRR method for all 400 events. Each value contains the event classification and the relative ratio. It was used an $FRR^{Threshold} = 0.35$.

| Subject ID | LSX | | | | LDX | | | | MSX | | | | MDX | | | |
|---|---|---|---|---|---|---|---|---|---|---|---|---|---|---|---|---|
| | 1 | 2 | 3 | 4 | 1 | 2 | 3 | 4 | 1 | 2 | 3 | 4 | 1 | 2 | 3 | 4 |
| 1 | TP-0.07 | TP-0.07 | TP-0.07 | TP-0.07 | TP-0.08 | TP-0.08 | TP-0.08 | TP-0.08 | TP-0.09 | TP-0.07 | TP-0.08 | TP-0.08 | TP-0.08 | TP-0.08 | TP-0.08 | TP-0.08 |
| 2 | TP-0.08 | TP-0.08 | TP-0.08 | TP-0.09 | TP-0.08 | TP-0.08 | TP-0.11 | TP-0.06 | TP-0.08 | TP-0.07 | TP-0.08 | TP-0.07 | TP-0.09 | TP-0.09 | TP-0.09 | TP-0.08 |
| 3 | TP-0.06 | TP-0.07 | TP-0.06 | TP-0.05 | TP-0.05 | TP-0.05 | TP-0.05 | TP-0.04 | TP-0.07 | TP-0.08 | TP-0.07 | TP-0.07 | TP-0.06 | TP-0.06 | TP-0.05 | TP-0.06 |
| 4 | TN-0.47 | TP-0.20 | TP-0.15 | TP-0.18 | TN-0.36 | TP-0.14 | TP-0.10 | TP-0.11 | TN-0.67 | TN-0.37 | FP-0.26 | FP-0.20 | TN-0.83 | FP-0.32 | TP-0.18 | TP-0.17 |
| 5 | TN-0.71 | TN-0.58 | TN-0.50 | TN-0.35 | TN-0.79 | TN-0.65 | TN-0.57 | TN-0.37 | TN-0.67 | TN-0.61 | TN-0.55 | TN-0.36 | TN-0.81 | TN-0.74 | TN-0.68 | TN-0.41 |
| 6 | TN-0.61 | TN-0.58 | TN-0.89 | TN-0.63 | TN-0.59 | TN-0.51 | TN-0.84 | TN-0.69 | TN-0.43 | TN-0.36 | TN-0.67 | TN-0.49 | TN-0.35 | FP-0.33 | TN-0.48 | TN-0.35 |
| 7 | TP-0.19 | TP-0.15 | TP-0.13 | TP-0.13 | TP-0.21 | TP-0.19 | TP-0.16 | TP-0.20 | TN-0.38 | TP-0.28 | TP-0.24 | TP-0.21 | TP-0.33 | TP-0.28 | TP-0.26 | TP-0.28 |
| 8 | TP-0.08 | TP-0.08 | TP-0.10 | TP-0.08 | TP-0.17 | TP-0.14 | TP-0.18 | TP-0.16 | TP-0.10 | TP-0.10 | TP-0.12 | TP-0.10 | TP-0.14 | TP-0.13 | TP-0.15 | TP-0.15 |
| 9 | FP-0.18 | TP-0.13 | TP-0.11 | TP-0.14 | FP-0.25 | TP-0.16 | TP-0.13 | TP-0.17 | TN-0.58 | TN-0.40 | TN-0.36 | TP-0.30 | TN-0.54 | TN-0.42 | TN-0.42 | TN-0.35 |
| 10 | TN-0.74 | TN-0.84 | TN-0.85 | TN-1.27 | TN-0.65 | TN-0.91 | TN-0.92 | TN-1.25 | TN-0.88 | TN-0.90 | TN-0.79 | TN-1.16 | TN-0.73 | TN-0.91 | TN-0.78 | TN-1.11 |
| 11 | TN-0.97 | TN-1.34 | TN-0.76 | TN-0.42 | TN-1.07 | TN-1.11 | TN-1.00 | TN-0.52 | TN-1.08 | TN-1.58 | TN-1.00 | TN-0.50 | TN-1.06 | TN-1.21 | TN-0.99 | TN-0.49 |
| 12 | TP-0.10 | TP-0.07 | TP-0.06 | TP-0.06 | TP-0.11 | TP-0.09 | TP-0.10 | TP-0.09 | TP-0.07 | TP-0.04 | TP-0.04 | TP-0.04 | TP-0.09 | TP-0.07 | TP-0.08 | TP-0.07 |
| 13 | TN-0.57 | TN-0.41 | TN-0.36 | TN-0.39 | TP-0.28 | TP-0.25 | TP-0.17 | TP-0.23 | TN-0.68 | TN-0.62 | TN-0.70 | TN-0.62 | TN-0.61 | TN-0.56 | TN-0.63 | TN-0.55 |
| 14 | TP-0.24 | TP-0.15 | TP-0.25 | FN-1.04 | TP-0.25 | TP-0.18 | TP-0.32 | FN-0.90 | TN-0.60 | TN-0.40 | TN-0.52 | TN-0.97 | TN-0.71 | TN-0.52 | TN-0.55 | TN-0.83 |
| 15 | TN-0.45 | TP-0.26 | TP-0.30 | TP-0.22 | TN-0.49 | TP-0.34 | TP-0.33 | TP-0.29 | TN-0.68 | TN-0.48 | TN-0.52 | TN-0.40 | TN-0.68 | TN-0.52 | TN-0.52 | TN-0.44 |
| 16 | TN-0.48 | TN-0.44 | FP-0.33 | FP-0.32 | TN-0.52 | TN-0.43 | TN-0.37 | TN-0.39 | TN-1.00 | TN-0.92 | TN-0.84 | TN-0.86 | TN-0.80 | TN-0.64 | TN-0.54 | TN-0.47 |
| 17 | TP-0.13 | TP-0.11 | TP-0.14 | TP-0.15 | TP-0.22 | TP-0.20 | TP-0.18 | TP-0.24 | TP-0.19 | TP-0.17 | TP-0.21 | TP-0.23 | TP-0.26 | TP-0.22 | TP-0.24 | TP-0.28 |
| 18 | TP-0.08 | TP-0.06 | TP-0.06 | TP-0.06 | TP-0.13 | TP-0.09 | TP-0.09 | TP-0.10 | TP-0.06 | TP-0.04 | TP-0.04 | TP-0.03 | TP-0.18 | TP-0.05 | TP-0.05 | TP-0.05 |
| 19 | TN-0.83 | TN-0.69 | TN-0.60 | TN-0.64 | TN-0.65 | TN-0.50 | TN-0.43 | TN-0.39 | TN-0.75 | TN-0.71 | TN-0.62 | TN-0.79 | TN-0.73 | TN-0.67 | TN-0.62 | TN-0.72 |
| 20 | TP-0.13 | TP-0.14 | TP-0.14 | TP-0.22 | TP-0.12 | TP-0.11 | TP-0.15 | TP-0.23 | TP-0.09 | TP-0.08 | TP-0.10 | TP-0.14 | TP-0.07 | TP-0.07 | TP-0.09 | TP-0.12 |
| 21 | TP-0.32 | TP-0.25 | TP-0.26 | TP-0.22 | TP-0.15 | TP-0.13 | TP-0.15 | TP-0.13 | TP-0.20 | TP-0.19 | TP-0.21 | TP-0.21 | TP-0.21 | TP-0.22 | TP-0.29 | TP-0.21 |
| 22 | TN-0.57 | TN-0.58 | TN-0.53 | FP-0.33 | TN-0.58 | TN-0.58 | TN-0.58 | FP-0.34 | TN-0.70 | TN-0.48 | TN-0.49 | TN-0.36 | TN-0.76 | TN-0.43 | TN-0.37 | FP-0.32 |
| 23 | TN-0.45 | TN-0.56 | TN-0.56 | TN-0.36 | TN-0.86 | TN-0.85 | TN-0.80 | TN-0.68 | TN-1.16 | TN-1.36 | TN-1.18 | TN-1.30 | TN-0.74 | TN-0.79 | TN-0.72 | TN-0.75 |
| 24 | TN-0.67 | TN-0.61 | TN-0.53 | TN-0.87 | TN-0.61 | TN-0.59 | TN-0.46 | TN-0.66 | TN-0.62 | TN-0.60 | TN-0.38 | TN-0.61 | TN-0.46 | TN-0.49 | TP-0.20 | FN-0.55 |
| 25 | TP-0.28 | TP-0.21 | TP-0.21 | TP-0.25 | TN-0.49 | TN-0.53 | TN-0.58 | TN-0.55 | TN-0.55 | TN-0.62 | TN-0.66 | TN-0.64 | FP-0.34 | FP-0.26 | FP-0.27 | FP-0.28 |

**Table 3.** Mean and standard deviation of the FRR, for Healthy and LBP subjects, obtained using the proposed method.

| GROUP | FRR |
| --- | --- |
| **HEALTHY** | $0.25 \pm 0.20$ |
| **LBP** | $0.55 \pm 0.26$ |

## 5. Discussion

The results of the VIS method (Table 1) and FRR method (Table 2) have shown that a subset of subjects exhibited FRP in all cycles and in all muscles, and it was indicated with $ID^{FRP}$ = *(ID1, ID2, ID3, ID8, ID12, ID17, ID18, ID20, ID21)*. The other subjects manifested FRP only in some cycles, some muscles or they didn't manifest FRP in any muscles. The reason for different patterns may lie in muscle fatigue [56], fear, or other features that vary from subject to subject (since there are many variables involved; the patient's report is useful for deepening the topic). Comparing the results of the VIS method with those obtained by the proposed algorithm is possible to carry out the performance evaluation, which is summarized in Table 2. Taking into account only the results related to the subgroup of healthy subjects indicated with $ID^{HEALTHY}$ = *(ID2, ID3, ID6, ID7, ID8, ID12, ID16, ID17, ID18, ID20, ID21, ID22, ID25)*. The proposed FRR algorithm shows no False Negative, so:

$$S_e^{HEALTHY} = \frac{TP}{P} = \frac{TP}{TP + FN} = \frac{147}{147 + 0} = 100\% \tag{5}$$

$$S_p^{HEALTHY} = \frac{TN}{N} = \frac{TN}{TN + FP} = \frac{51}{51 + 10} = 84\% \tag{6}$$

This means that the ability of the FRR method, Equation (1), to discriminate FRP presence, compared to the VIS method, in the healthy subgroup (with a total number of 147 events) is equal to 100% (Equation (5)). In the literature, Alison et al. [20] reported a sensibility of 100% evaluating other types of FRR methods on 24 events collected in an acquisition campaign that involved only healthy subjects. Therefore, we have obtained similar performance comparing our results with those reported by Alison in the same type of subset but using a greater number of events. While as reported in Equation (6), the proposed FRR algorithm is less performing to discriminate FRP absence in the healthy subgroup (with a total number of 61 events).

We can do similar considerations taking into account only the results related to the subgroup of LBP subjects indicated with $ID^{LBP}$ = *(ID1, ID4, ID5, ID9, ID10, ID11, ID13, ID14, ID15, ID19, ID23, ID24)*. The proposed algorithm shows:

$$S_e^{LBP} = \frac{TP}{P} = \frac{TP}{TP + FN} = \frac{48}{48 + 3} = 94.1\% \tag{7}$$

$$S_p^{LBP} = \frac{TN}{N} = \frac{TN}{TN + FP} = \frac{136}{136 + 5} = 96.5\% \tag{8}$$

This means that the ability of the FRR method, Equation (1), to discriminate FRP presence, compared to the VIS method, in LBP subgroup (with a total number of 51 events) is equal to 94.1% (Equation (7)). While as reported in Equation (8), the proposed FRR algorithm is more performing to discriminate FRP absence in the LBP subgroup (with a total number of 141 events).

Summarizing, in this study we have extended the evaluation of the FRP taking subjects healthy and with LBP, producing 400 events, and evaluating the performances of the proposed algorithm on the entire group of subjects. The results are illustrated in Table 2, and the performances are indicated in Equations (2)–(4). The table shows that this algorithm correctly recognized 195 of the 198 events with FRP and 187 of the 202 events without FRP. Only 15 False Positives and three False Negatives occurred on a total of 400 events. FNs can occur in events where the end of a phase does not exactly coincide with the end of the sEMG pattern for that phase. Then the sEMG pattern excess enters in the

new phase and it alters the average value, changing the FRR. FPs can occur in events where the FRP is absent and the sEMG activity is moderate and very variable during the full flexion phase. In these cases, the FRR value does not exceed the nominal threshold as the mean value during the full flexion phase is not big enough compared to the activity during the extension.

As shown in equations Equations (2)–(4), the algorithm has therefore accuracy of 95.5%, a sensitivity of 98.5%, and a specificity of 92.6%. Comparing sensitivity Equation (3) and specificity Equation (4) results is clear that the algorithm discriminates very well subjects with FRP while it's more difficult to identify subjects without FRP. Furthermore, as reported in Equation (6), this reduced specificity is mainly caused by healthy subjects who have not FRP.

Most cases of FPs had a value very near to the threshold level ($FRR^{Threshold}$) used to make the decision; often, when the $FRR_i^C$ value is near the threshold it can take a different decision compared to the VIS method (the closer FRR gets to the threshold, the greater the uncertainty of the decision), which makes the detection a non-trivial problem.

## 6. Conclusions

This paper deepens the investigation of FRP on back muscles using a WBSN composed of four sEMG sensors and a wearable device that integrates accelerometer, gyroscope, and magnetometer. The raw data collected from the WBSN during the flexion-relaxation test are processed by an algorithm able to identify the phases of which the test is composed, provide an evaluation of the myoelectric activity and automatically detect the FRP presence/absence. The proposed algorithm was tested using the data acquired in an acquisition campaign conducted to evaluate the flexion-relaxation phenomenon on the back muscles of subjects with and without LBP. The computed signal, identifying the phases of the flexion relaxation test, represented very well the subject's trunk real motion. Moreover, the phases signal trend varies in correspondence of the angular variations causing the activation or not of the muscles. The assessment of the myoelectric activity on back muscles provided by the proposed algorithm was evaluated by the medical staff as a useful tool to identify and cluster different patterns, visually analyse FRP presence/absence with the VIS method and aid the clinical assessment of the FRP. The ratio, expressed by Equation (1), was computed for each event using the data collected in the acquisition campaign. This FRR parameter is then compared with an empirical threshold value and the final decision about flexion-relaxation phenomenon presence/absence is taken. The threshold level used in the algorithm to detect FRP seemed to classify very well the events collected in the dataset. Indeed, the results show that the proposed algorithm for the FRP detection obtained an accuracy of 95.5%, a sensitivity of 98.5%, and a specificity of 92.6%, processing the data acquired from the subjects with and without LBP. Despite the excellent results achieved, future developments will concern the planning of a new acquisition campaign and the study of new solutions able to detect the FRP and to improve the performance of the proposed algorithm. In the future acquisition campaign, different motion tasks will be taken into consideration to evaluate the FRP and the psycho-physical conditions of the subjects involved (health conditions, level of stress, etc.) will be carefully monitored before carrying them out. Moreover, the relationship between the FRR method and the low back pain will be studied in future works in order to try to discriminate healthy subjects from LBP patients by analysing FRRs parameters without knowing a priori the clinical conditions.

**Author Contributions:** Conceptualization, M.P., A.B., L.P. and P.P.; methodology, M.P. and P.P.; software, M.P. and A.B.; validation, M.P. and L.P.; formal analysis, M.P. and M.V.; investigation, M.P.; resources M.P., P.P. and M.V.; data curation, M.P.; writing–original draft preparation, M.P. and A.B.; writing–review and editing, M.P. and A.B.; visualization, M.P and A.B; supervision, P.P.; project administration, P.P.; funding acquisition, P.P. All authors have read and agreed to the published version of the manuscript.

**Funding:** This research was funded by Department of Information Engineering (DII), Università Politecnica delle Marche, project title "A network-based approach to uniformly extract knowledge and support decision making in heterogeneous application contexts", grant number RSA-B2018.

**Acknowledgments:** The authors are grateful to the Istituto di Riabilitazione Santo Stefano (www.sstefano.it) for the opportunity to carry out this research work and the technical support of the materials used for experiments.

**Conflicts of Interest:** The authors declare no conflict of interest.

## Abbreviations

The following abbreviations are used in this manuscript:

| | |
|---|---|
| ACC | Acceleration |
| BSN | Body Sensor Network |
| ECG | Electrocardiogram |
| FN | False Negative |
| FP | False Positive |
| FER | Flexion Extension Ratio |
| FRP | Flexion Relaxation Phenomenon |
| FRR (or plural FRRs) | Flexion Relaxation Ratio (s) |
| GYR | Gyroscope |
| ID | Identification Number |
| LBP | Low Back Pain |
| LDX | Longissimus Right |
| LSX | Longissimus Left |
| MAG | Magnetic field |
| MDX | Multifidus Right |
| MSX | Multifidus Left |
| NRS-11 | Numeric Rating Scale |
| RMS | Root Mean Square |
| sEMG | Surface Electromyography |
| TP | True Positive |
| TN | True Negative |
| VIS | Visual Inspection |
| WBSN | Wireless Body Sensor Network |

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
