# Peer review of "A Wireless Body Sensor Network for Clinical Assessment of the Flexion-Relaxation Phenomenon"

_electronics, doi:10.3390/electronics9061044_

Round 1

Reviewer 1 Report

As far as I see, You have chosen only one sort of exercise, do you think, if you extend your study for different exercises OR poses, your study will be more useful moreover, adopt-able. 

Also, it would be great if you examine the stress level for each subject performing the tests for example through EEG.

There is a sensor device called shadow motion (I guess). How different your hardware is from that. What is the advantage of your setup as compared to that sensor setup.

Please follow these article for more information.

Gait Analysis Using Shadow Motion, P Kumar et al., 2017 4th IAPR Asian Conference on Pattern Recognition (ACPR), 453-458

Multimodal gait recognition with inertial sensor data and video using evolutionary algorithm, P Kumar et al., IEEE Transactions on Fuzzy Systems 27 (5), 956-965

Author Response

Comment R.1.1: As far as I see, You have chosen only one sort of exercise, do you think, if you extend your study for different exercises OR poses, your study will be more useful moreover, adopt-able.

Also, it would be great if you examine the stress level for each subject performing the tests for example through EEG.

Response: We thank the reviewer for the provided comments that have allowed us to clarify some concepts. It is the main exercise commonly used by the scientific community to evaluate the FRP. This type of exercise was mainly used in references [2], [23] and so on. However, the use of different exercises or poses in a new acquisition campaign could provide additional material to further demonstrate the effectiveness of the proposed system. Due to the Covid-19 lockdown, the acquisition campaign was stopped and the results of it were presented in this paper. Given the excellent performances obtained, the rehabilitation institute contributing to this study has decided to undertake a new acquisition campaign in which it will be possible to extend the study for different exercises and poses as well as monitor the stress level of the subjects who perform them.

Action(s) taken: Following the reviewer's suggestion, the conclusion section has been reworked adding some remarks relating to future works.

Comment R.1.2: There is a sensor device called shadow motion (I guess). How different your hardware is from that. What is the advantage of your setup as compared to that sensor setup.

Please follow these articles for more information.

Gait Analysis Using Shadow Motion, P Kumar et al., 2017 4th IAPR Asian Conference on Pattern Recognition (ACPR), 453-458

Multimodal gait recognition with inertial sensor data and video using evolutionary algorithm, P Kumar et al., IEEE Transactions on Fuzzy Systems 27 (5), 956-965

Response: Thank the reviewer for the remark. The hardware of the proposed wearable device is the same as that used in one node of the shadow motion system. The sensor setup of the shadow motion system is used for gait analysis. The proposed setup is used to estimate the subject’s inclination and identify the phases of which the test is composed. Based on your suggestion, the aforementioned works have been included in the references for our work giving their importance and relatedness.

Action(s) taken: The two contributions have been inserted in the introduction section where we navigate existing literature contributions in our field of interest.

Attached is the paper with the corrections made (in red).

Reviewer 2 Report

The authors presented an interesting study to valuation of the flexion-relaxation phenomenon using a wireless body sensor network consisting of sEMG sensors in association with a wearable device that integrates accelerometer, gyroscope, and magnetometer. In fact, they evaluated the performance of a Wireless Body Sensor Network (WBSN) able to analyze, quantitatively and objectively, the surface electromyography of the low back muscles (longissimus and multifidus) and automatically detect FRP presence/absence in both healthy subjects and patients. To this reviewer, the topic is interesting however there are some major limitations. The authors need to address them before publishing their interesting study.

Comments:

1-The originality of this study is questionable. The authors should clearly explain the difference of their goals with other studies who classified healthy subjects vs. patients with low back pain.

2-The authors should justify why they focused on Flexion-Relaxation Phenomenon among many evaluation tools and methods. What is the advantage of this method vs. others? They need to review the literature for other methods and explain why they selected Flexion-Relaxation Phenomenon.

3- There are many wearable devices including IMU and textile sensor for measuring the angle and detect the activities of upper body especially trunk movement. The authors should explain why they chose this type of wearable system for measuring the kinematic data vs other devices for example textile sensors. Also, they need to evaluate the usability. Please see the below papers for example:

  • Mukhopadhyay, Subhas Chandr;Wearable sensors for human activity monitoring: A review; IEEE sensors journal 15.3 (2015): 1321-1330.
  • Mokhlespour Esfahani, Mohammad, and Maury Nussbaum. “Preferred placement and usability of a smart textile system vs. inertial measurement units for activity monitoring.”; Sensors 18.8 (2018): 2501.
  • Bergmann, Jeroen, Vikesh Chandaria, and Alison McGregor. "Wearable and implantable sensors: the patient’s perspective." Sensors 12.12 (2012): 16695-16709.
  • Bergmann, J. H. M., and A. H. McGregor. "Body-worn sensor design: what do patients and clinicians want?." Annals of biomedical engineering 39.9 (2011): 2299-2312.

4-Lines 168-169: The author needs to explain how increasing the sample frequency almost 10 times did not impact the original data. This may be a significant issue for the original data. “the inclination signal was resampled from 128 Hz to 2kHz using a linear interpolation function”.

5-Line 176: “an empirical threshold level equal to 0.09”. The author should explain how they find this amount. At least they should cite another study.

6- I recommend the author to describe the participants including healthy subject and patients in a separate section. They need to report the anthropometric data of their participants clearly.

7-In Discussion, the authors need to mention the limitations of their study. Also, they have to compare their finding with other methods for discriminating between healthy subject and patients. There are many studies that they classified healthy subject vs. patients with low back pain.

Author Response

Comment R.2.1: The originality of this study is questionable. The authors should clearly explain the difference of their goals with other studies who classified healthy subjects vs. patients with low back pain.

Response: We thank the reviewer for the opportunity to improve the quality of the paper. We decided to proceed with a better explanation by adding what is present in the literature. With our study, we don't want to provide a method to classify healthy subjects vs. patients. The main purpose of the paper is to provide a useful tool for the automatic identification of the FRP in both healthy and LBP subjects. In the literature, several studies have proposed systems for the evaluation of FRP only on healthy subjects [18] [22] [23]. In this study, starting from the data acquired by the WBSN, an algorithm based on the FRR method was implemented to automatically detect the FRP even in LBP subjects. Recent studies have proposed a system for assessment of the FRP in both healthy subjects and LPB subjects. In particular, Ducina et al. have used a 12-camera motion analysis system to determine the inclination [40]. Other recent studies have started to use wearable wireless inertial systems to study the FRP [41] [42]. However, these studies have not proposed automatic algorithms for FRP detection or at least have not provided the results obtained with them.

Action(s) taken: A detailed explanation of this remark has been added inside the introduction section.

Comment R.2.2: The authors should justify why they focused on Flexion-Relaxation Phenomenon among many evaluation tools and methods. What is the advantage of this method vs. others? They need to review the literature for other methods and explain why they selected Flexion-Relaxation Phenomenon.

Response: Our work was born from a collaboration with a rehabilitation institute that needs a tool for automatic FRP detection. The rehabilitation institute is studying FRP in rehabilitation treatments and it focused the attention on the FRP because is known that the pain interferes with both afferent and efferent aspects of neuromuscular control [4] [5] [6]. To evaluate if the subject has normal neuromuscular patterns amongst the various physiological indicators of LBP, the FRP has been one of the most studied surface electromyographic responses in literature [1] [3] [7] [8] [9].

Action(s) taken: Some aspects related to these remarks has been added inside the introduction section.

Comment R.2.3: There are many wearable devices including IMU and textile sensor for measuring the angle and detect the activities of upper body especially trunk movement. The authors should explain why they chose this type of wearable system for measuring the kinematic data vs other devices for example textile sensors. Also, they need to evaluate the usability. Please see the below papers for example:

  • Mukhopadhyay, Subhas Chandr;Wearable sensors for human activity monitoring: A review; IEEE sensors journal 15.3 (2015): 1321-1330.
  • Mokhlespour Esfahani, Mohammad, and Maury Nussbaum. “Preferred placement and usability of a smart textile system vs. inertial measurement units for activity monitoring.”; Sensors 18.8 (2018): 2501.
  • Bergmann, Jeroen, Vikesh Chandaria, and Alison McGregor. "Wearable and implantable sensors: the patient’s perspective." Sensors 12.12 (2012): 16695-16709.
  • Bergmann, J. H. M., and A. H. McGregor. "Body-worn sensor design: what do patients and clinicians want?." Annals of biomedical engineering 39.9 (2011): 2299-2312.

Response: We have chosen to use the IMU as it represents the best system for our purposes (FRP evaluation) as it can be perfectly integrated with the sEMG system. The Smart Textile System certainly represents an interesting idea for future developments.

Action(s) taken: Based on your suggestion, some aforementioned works have been included in the references for our work giving their importance and relatedness. We have added in our paper the second reference to add alternative solutions for activity monitoring.

Comment R.2.4: Lines 168-169: The author needs to explain how increasing the sample frequency almost 10 times did not impact the original data. This may be a significant issue for the original data. “the inclination signal was resampled from 128 Hz to 2kHz using a linear interpolation function”.

Response: For gait analysis applications, typical sampling rates are between 50 and 100 Hz [van der Kruk, Eline, and Marco M. Reijne. "Accuracy of human motion capture systems for sport applications; state-of-the-art review." European journal of sport science 18.6 (2018): 806-819.]. During the flexion-extension tests, the rapidity of the subject’s movements is much slower than that involved in gait analysis therefore the sampling frequency used (128 Hz) is high enough to fully describe the phenomenon.

Therefore, an increase in frequency, albeit fictitious through oversampling, is not significant. This ploy was used only to align the sampling instants of the two systems. In this way, the sEMG signal and the inclination signal can be represented in the figures shown in the paper with the same reference time.

Action(s) taken: An explanation of this remark has been added inside the “Algorithm for FRP clinical assessment” section.

Comment R.2.5: Line 176: “an empirical threshold level equal to 0.09”. The author should explain how they find this amount. At least they should cite another study.

Response: We thank the reviewer for the provided comments that have allowed us to clarify some concepts. We have obtained this threshold value empirically by carrying out a series of tests. In particular, a different dataset consisting acquisitions on completely healthy subjects was analyzed by superimposing sEMG patterns on the “Inclination” signal and the “Processed angular velocity” signal (see Fig. 4). The chosen threshold represents the best value able to identify the various phases since the sEMG patterns with that value coincide perfectly with the duration of the phases.

Action(s) taken: An explanation of how we found the threshold was made has been added inside the “Algorithm for FRP clinical assessment” section.

Comment R.2.6 I recommend the author to describe the participants including healthy subject and patients in a separate section. They need to report the anthropometric data of their participants clearly.

Response: We thank the reviewer for the opportunity to improve the quality of the paper. Based on your comment, additional data have been added to highlight the subject’s characteristics. Further details on the participants are also available in the data paper describing the experiment and how it was carried out [43].

Action(s) taken: For greater clarity, Table 1 was modified by adding the group to which the subjects belonged (healthy and with LBP).

Comment R.2.7 In Discussion, the authors need to mention the limitations of their study. Also, they have to compare their finding with other methods for discriminating between healthy subject and patients. There are many studies that they classified healthy subject vs. patients with low back pain.

Response: In literature, many studies have focused on the identification of healthy and LBP subjects but in this study, a tool able to accurately identify the FRP has been proposed. Future work will be focused on to the evaluation of the correlation between the presence/absence of FRP and presence/absence of LBP. The aim of this work is to propose a useful tool for the automatic identification of the FRP in both healthy subject and LBP. In the Discussion section, a comparison with a study proposing algorithms for the detection of FRP only in healthy subjects has been discussed.

Action(s) taken: These aspects have been better clarified in the introduction and conclusion sections.

Attached is the paper with the corrections made (in red).

Reviewer 3 Report

This paper presents an interesting application of Wireless Body Sensor Networks (WBSN) for Flexion-Relaxation Phenomenon (FRP) detection. The manuscript is well written and could be suitable to be published in the Electronics journal provided that the following comments are implemented within the document:

- ¿Is the use of WBSN for the detection of FRP presence/absence an innovative aspect of this paper? If not, other related works should be mentioned specifying the new contribution of this manuscript. If WBSN have never been employed for FRP detection before, it should be clearly stated in the Introduction section.
- From Table 2, it seems that FPs and FNs are often grouped within the same subjects. For example, IDs 4, 9, 14, 16, 22 and 25. Could you identify specific characteristics in those subjects to explain such fact?
- A Table with the statistics of the subjects (sex, age, weight, etc.) is needed.
- The graphics of Fig. 6 should be larger in order to clearly see the details.
- In Table 1, I would select the letter 'A' for Absence or else keep the criterion mentioned within the text: 'P' for Positive and 'N' for Negative.

Author Response

Comment R.3.1: Is the use of WBSN for the detection of FRP presence/absence an innovative aspect of this paper? If not, other related works should be mentioned specifying the new contribution of this manuscript. If WBSN have never been employed for FRP detection before, it should be clearly stated in the Introduction section.

Response: We thank the reviewer for the opportunity to improve the quality of the paper. We decided to proceed with a better explanation by adding what is present in the literature.

Several studies have proposed systems for the evaluation of FRP only on healthy subjects. In particular, Ritvanen et al. [22] have used one sEMG system while Alison Schinkel-Ivy et al. [18] have added a motion capture system to it. Sihvonen et al. [23] have proposed a similar system based on a wired Body Sensor Network. In this study, we wish to evaluate the performance of Wireless Body Sensor Network (WBSN) able to evaluate the FRP in both healthy subjects and patients with LBP. Recent studies have proposed a system for assessment of the FRP even in LPB subjects. In particular, Ducina et al. have used a 12-camera motion analysis system to determine the inclination [40]. Other recent studies have started to use wearable wireless inertial systems to study the FRP [41] [42]. However, these studies have not proposed automatic algorithms for detecting FRP or at least have not provided the results obtained with them. In this study, starting from the data acquired by the WBSN, an algorithm based on the FRR method was implemented to automatically detect the FRP.

Action(s) taken: A detailed explanation of these remarks has been added inside the introduction section.

Comment R.3.2: From Table 2, it seems that FPs and FNs are often grouped within the same subjects. For example, IDs 4, 9, 14, 16, 22 and 25. Could you identify specific characteristics in those subjects to explain such fact?

Response: We thank the reviewer for the provided comments that have allowed us to clarify some concepts. As shown in the reworked Table 1, it is not possible to identify specific characteristics in the indicated subjects. FPs and FN are due to the features of the acquired signals. In fact, FNs can occur in events where the end of a phase does not exactly coincide with the end of the sEMG pattern for that phase. Then the sEMG pattern excess enters in the new phase and it alters the average value, changing the FRR. Moreover, FPs can occur in events where the FRP is absent and the sEMG activity is moderate and very variable during the full flexion phase. In these cases, the FRR value does not exceed the nominal threshold as the mean value during the full flexion phase is not big enough compared to the activity during the extension. Furthermore, to give greater emphasis to the results relating to the method used, we have included Table 3.

Action(s) taken: A detailed explanation of why FN and FP occurred has been added inside the discussion section. Moreover, Table 3 has benne added inside the results section in order to better explain the obtained results.

Comment R.3.3: A Table with the statistics of the subjects (sex, age, weight, etc.) is needed

Response: We thank the reviewer for the opportunity to improve the quality of the paper. Based on your comment, additional information have been added to highlight the subjects’ characteristics.

Action(s) taken: Significant statistics of the subjects have been added to Table 1.

Comment R.3.4: The graphics of Fig. 6 should be larger in order to clearly see the details.

Response: Thank the reviewer for the remark. The suggestions have been faithfully followed.

Action(s) taken: In order to clearly see the details, the previous Fig. 6 has been divided into two figures. To unify the figures, the same thing has also been done for the Fig. 7.

Comment R.3.5: In Table 1, I would select the letter 'A' for Absence or else keep the criterion mentioned within the text: 'P' for Positive and 'N' for Negative.

Response: Thank the reviewer for the remark. The suggestions have been faithfully followed.

Action(s) taken: The caption of Table 1 has been reworked using ‘P' for Positive and 'N' for Negative.

Attached is the paper with the corrections made (in red).

Round 2

Reviewer 1 Report

my comments are answered. 

thanks

Reviewer 2 Report

Thank you for providing detailed responses to my comments. You addressed all of my concerns.